# Differential expression of MAGEA6 toggles autophagy to promote pancreatic cancer progression

Yiu Huen Tsang[1,2]*, Yumeng Wang[3], Kathleen Kong[1], Caitlin Grzeskowiak[1], Oksana Zagorodna[1], Turgut Dogruluk[1], Hengyu Lu[1], Nicole Villafane[1,4], Venkata Hemanjani Bhavana[1], Daniela Moreno[1], Sarah H Elsea[1], Han Liang[3,5], Gordon B Mills[2,5], Kenneth L Scott[1†]

[1]Department of Molecular and Human Genetics, Baylor College of Medicine, Houston, United States; [2]Cell, Develop & Cancer Biology, Oregon Health & Science University, Portland, United States; [3]Department of Bioinformatics and Computational Biology, The University of Texas MD Anderson Cancer Center, Houston, United States; [4]Michael E DeBakey Department of Surgery, Baylor College of Medicine, Houston, United States; [5]Department of Systems Biology, The University of Texas MD Anderson Cancer Center, Houston, United States

*For correspondence:
tsangsa@ohsu.edu

†Deceased

Competing interests: The authors declare that no competing interests exist.

**Abstract** The melanoma-associated antigen family A (MAGEA) antigens are expressed in a wide variety of malignant tumors but not in adult somatic cells, rendering them attractive targets for cancer immunotherapy. Here we show that a number of cancer-associated MAGEA mutants that undergo proteasome-dependent degradation in vitro could negatively impact their utility as immunotherapeutic targets. Importantly, in pancreatic ductal adenocarcinoma cell models, MAGEA6 suppresses macroautophagy (autophagy). The inhibition of autophagy is released upon MAGEA6 degradation, which can be induced by nutrient deficiency or by acquisition of cancer-associated mutations. Using xenograft mouse models, we demonstrated that inhibition of autophagy is critical for tumor initiation whereas reinstitution of autophagy as a consequence of MAGEA6 degradation contributes to tumor progression. These findings could inform cancer immunotherapeutic strategies for targeting MAGEA antigens and provide mechanistic insight into the divergent roles of *MAGEA6* during pancreatic cancer initiation and progression.

## Introduction

Cancer/testis antigens (CTAs) are encoded by genes whose expression is normally restricted to male germ cells but is de-repressed in various human cancers (*Chen et al., 2017*; *Hofmann et al., 2008*). Their unique expression patterns could have utility in cancer diagnosis, prognosis, and immunotherapy. The MAGE protein family of CTAs, including the MAGEA, B, and C Type one subfamily members, (*Barker and Salehi, 2002*; *Simpson et al., 2005*) are of particular interest. The aberrant expression of Type 1 MAGEs in cancer is a result of promoter hypomethylation due to genome-wide epigenetic reprogramming. The high tumor specificity of MAGEA expression led to multiple clinical trials targeting MAGEA genes with immunotherapy agents (*Vansteenkiste et al., 2013*; *Zajac et al., 2017*). However, responses have been modest (*Connerotte et al., 2008*).

MAGEA genes have been proposed to function as oncogenes; however, the underlying mechanisms are poorly understood (*Weon and Potts, 2015*). Consistent with a role in tumor progression, high MAGEA protein expression has been associated with advanced stage (*Liu et al., 2008*; *Sang et al., 2017*; *Wang et al., 2016*) and poor prognosis (*Weon and Potts, 2015*; *Xu et al., 2014*; *Zhai et al., 2016*; *Zhang et al., 2015*) in multiple cancer lineages. Interactions between MAGEs and

really interesting new gene (RING) domain containing E3 ubiquitin ligases (*Doyle et al., 2010*) enhance E3 ubiquitin ligase activity toward downstream substrates and lead to their degradation (*Doyle et al., 2010*), which may contribute to the oncogenic activity of MAGEs. A recent proteomic study revealed a negative association between autophagy activators and MAGEA protein expression in human melanoma (*Shukla et al., 2018*). In addition, MAGEA3 and MAGEA6 suppress autophagy in lung and colorectal cancer models (*Pineda et al., 2015*). The effects of MAGEA on autophagy provide a potential mechanism for oncogenic effects of MAGEA.

Autophagy promotes cell survival by sustaining cellular synthetic pathways and energy homeostasis under metabolic stress. It also preserves long-term cellular integrity by preventing the accumulation of misfolded proteins and malfunctioning organelles (*White, 2012*; *Yang and Klionsky, 2010*). Both pro- and anti-tumorigenic properties of autophagy have been reported and are largely determined by specific tumor stage and genetic context (*Guo et al., 2011*; *Guo et al., 2013*; *Lock et al., 2011*; *Rao et al., 2014*; *White, 2012*). During early steps in tumor development, autophagy acts as a tumor suppressor by facilitating DNA damage repair and reducing reactive oxygen species and thus maintaining genome integrity (*Panda et al., 2015*). At late stages of tumor development, autophagy increases the ability of tumor cells to cope with endogenous stress and increases chemotherapy or radiotherapy resistance (*Pylayeva-Gupta et al., 2011*; *White, 2012*) thus promoting tumorigenic behavior. Oncogenic *KRAS* mutations, which are present in more than 90% of pancreatic ductal adenocarcinoma (PDAC) tumors, represent the earliest driving event for PDAC (*Almoguera et al., 1988*). Knockdown of core members of the autophagy initiation complex in $Kras^{G12D/+}$ transgenic mice increased accumulation of low-grade, pre-malignant pancreatic intraepithelial neoplasia lesions but rendered these lesions resistant to progression into invasive PDAC (*Rosenfeldt et al., 2013*; *Yang et al., 2014*). In contrast, pharmacologic inhibition of autophagy in established PDAC patient-derived xenografts induced apoptosis and reduced proliferation (*Yang and Klionsky, 2010*). Thus, autophagy appears to play a negative role in PDAC initiation but is important for late-stage tumor development.

Given the complex role of autophagy in PDAC, the role of interactions between *MAGEA6*, which has been found to be aberrant in PDAC, and the autophagy machinery in PDAC initiation remains unclear. In addition, there is an urgent need to understand the impact of cancer-specific mutations in *MAGEA6* on tumor aggressiveness and on immunotherapy strategies (*Caballero et al., 2010*; *Hagiwara et al., 2016*). We therefore developed a comprehensive MAGEA cancer-specific mutation series to interrogate the impact of these mutations on protein expression and function. We also examined the role of PDAC-specific *MAGEA6* variants using in vitro and in vivo PDAC models.

## Results

### Mutational landscape of MAGEA genes in cancer

The MAGEA gene family is located on the X chromosome and consists of thirteen protein-encoding genes (*MAGEA1* to *A6*, *A8* to *A12*, *A2B*, and *A9B*) and one pseudogene, *MAGEA7P*. About 1000 unique non-silent MAGEA gene aberrations have been identified by ICGC (*ICGC/TCGA Pan-Cancer Analysis of Whole Genomes Consortium, 2020*), The Cancer Genome Atlas (TCGA), and Catalogue of Somatic Mutations in Cancer (COSMIC) (*Tate et al., 2019*) pan-cancer analyses (*Figure 1A*). Most of these gene aberrations are missense mutations, and they are clustered on *MAGEA1*, *A3*, *A4*, *A6*, *A8*, *A10*, *A11*, and *A12*. As a group, MAGEA gene family members, are more frequently mutated in TCGA lung cancer patients and pancreatic cancer patients when compared to a randomized gene set of 1000 genes with similar size to MAGEAs (*Figure 1B*), suggesting that mutation of MAGEA genes is functionally important in cancer.

MAGEA genes represent an attractive group of antigens for cancer immunotherapy owing to their upregulation in cancer tissues. Indeed, over the past 10 years, 42 phase I and II clinical trials (*Supplementary file 2*) have explored MAGEA3, A4, A6, A10, and A12 as targets for immunotherapy. More than 500 recurrent mutations have been identified in the five clinically studied MAGEA genes in TCGA, ICGC, and COSMIC pan-cancer analyses (*Figure 1C*). These mutations were uniformly distributed along the entire gene without any obvious mutational hotspots.

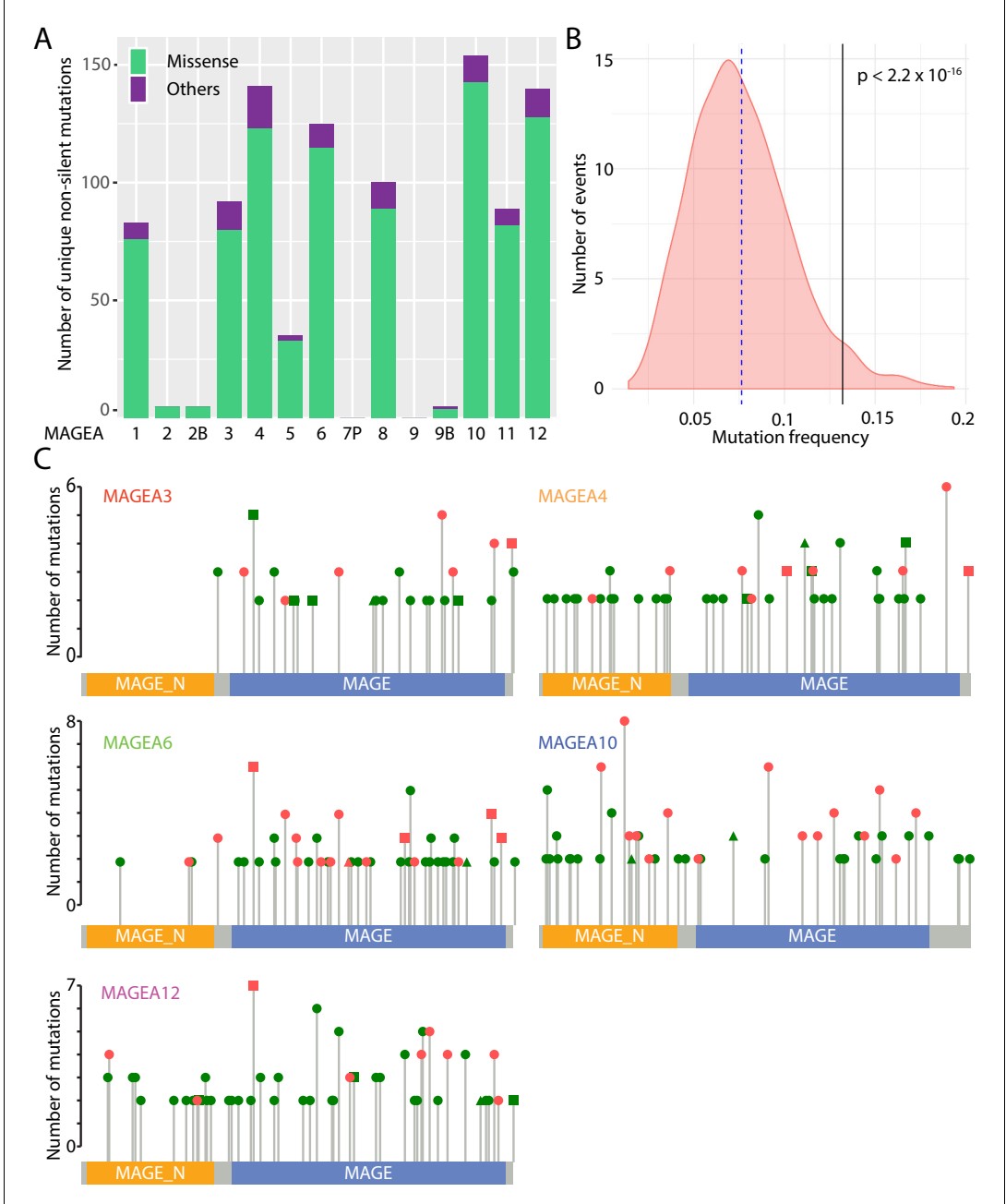

**Figure 1.** Mutational landscape of MAGEA genes in pan-cancer study. (**A**) Histogram of missense mutations and others (nonsense, frameshift, in-frame indels, and splice site mutations) across the entire MAGEA family. (**B**) Mutation frequency analysis of MAGEA genes. Pink area represents the mutation frequency distribution of 1000 randomly selected genes with sizes similar to MAGEA genes. Blue dotted line and black line indicate the mean mutation frequency of the 1000 genes and the MAGEA gene respectively. P value was calculated by a one-sample Wilcoxon test. (**C**) Lollipop diagrams indicate the distribution and number of recurrent mutations in individual MAGEA genes. Reported amino acid changes: missense mutations (dot), others (triangle), and both (square) are marked in pink for those selected for immunoblot analysis in *Figure 2*.

## Cancer-specific mutations of MAGEA genes reduce their protein expression

We leveraged our High-Throughput Mutagenesis and Molecular Barcoding (HiTMMoB) technology (*Tsang et al., 2016*) to mimic 82 recurrent mutation events (found in at least 150 cancer patients) in the five MAGEA genes that have been evaluated in clinical trials (*Figure 1C* and *Supplementary file 3*). To evaluate the impact of the mutations on protein expression, equal amounts of mutant

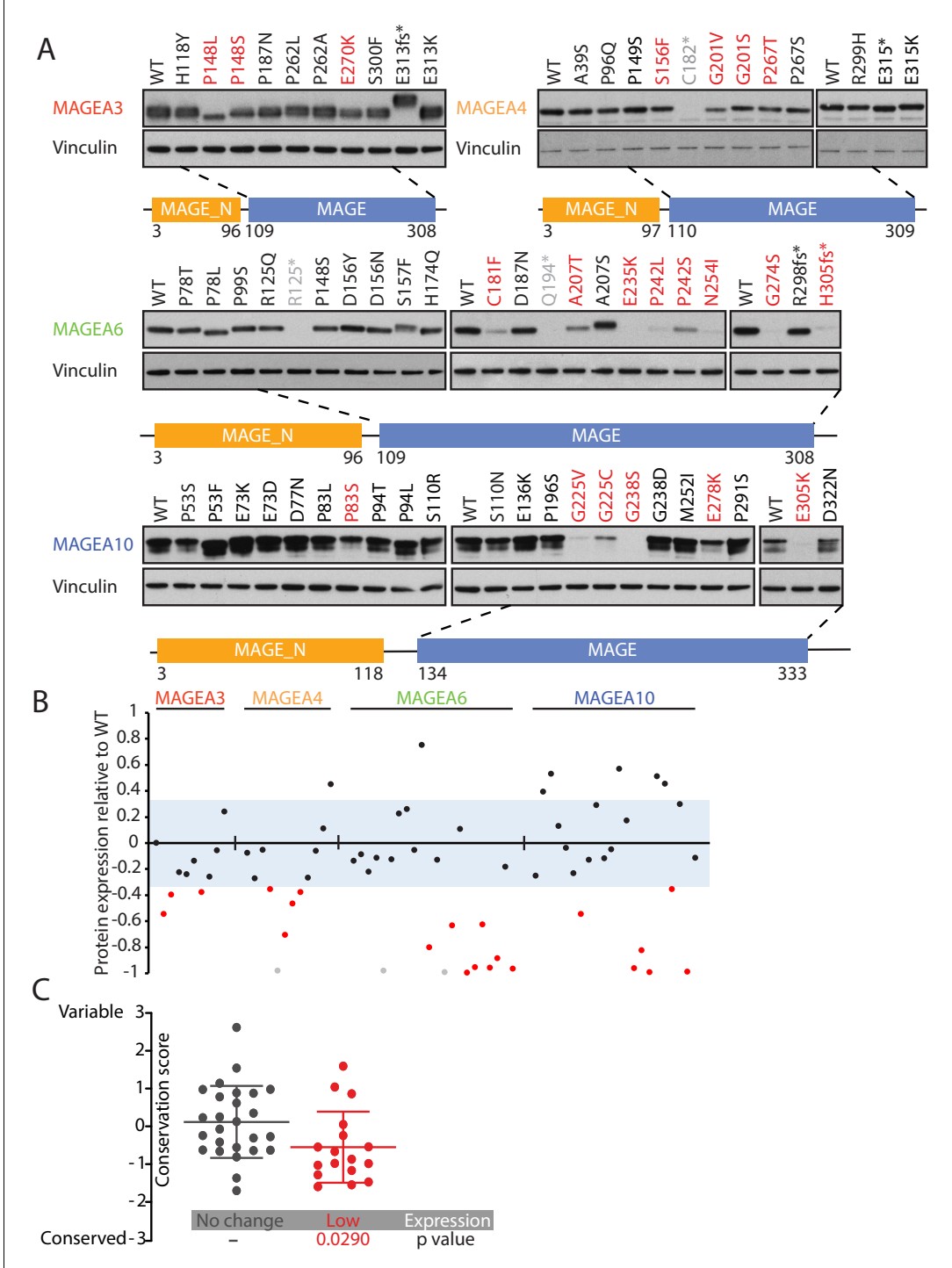

**Figure 2.** Cancer-specific mutations of *MAGEA* genes reduce their protein expression. (A) Immunoblot analysis of MAGEA3, A4, A6 and A10 variants expressed in HEK293T cells. Variants that are not recognized by the antibody (gray), variants expressed at levels 33% (three standard deviation determined by the expression deviation analysis in *Figure 2—figure supplement 2*) or less than those of the wild-type (WT) (red) are indicated. MAGE_N: N-terminal MAGE domain. (B) Densitometry analysis of protein expression of the MAGEA variants in (A). Each variant is represented by a dot, and the dots are shown in the same order and colors as in (A). The blue area represents three standard deviations, determined from *Figure 2—figure supplement 2*. (C) Conservation score analysis (mean ± standard deviation) of amino acids that show reduced protein expression and those that show unchanged protein expression when mutated. P value was calculated by two-tailed unpaired t-test (N = 26 for no change cohort, N = 17 for low-expression cohort).

The online version of this article includes the following figure supplement(s) for figure 2:

*Figure 2 continued on next page*

*Figure 2 continued*

**Figure supplement 1.** Immunoblot analysis of MAGEA12 variants expressed in HEK293T cells.
**Figure supplement 2.** Immunoblot (top) and densitometry plot (bottom) of the expression deviation analysis.
**Figure supplement 3.** Evolutionary conservation study of MAGEAs.

plasmids were transfected individually into HEK293T cells. Due to the specificity limit of the antibody targeting MAGEA12, we were unable to confirm protein level changes of MAGEA12 variants (*Figure 2—figure supplement 1*), and thus we decided to remove them from the statistical analysis. About one-third of the MAGEA variants (21/67; 20 missense mutations and one nonsense mutation) showed a marked reduction in protein expression compared with their wild-type counterparts (*Figure 2A and B* and *Figure 2—figure supplement 2*). For most of the under-expressed variants (20/21), the mutations lay within a highly conserved MAGE homology domain (MHD) domain (*Figure 2—figure supplement 3*) at the C-terminus of the four MAGEA genes (*Figure 2A*). The amino acids that led to lower protein levels when mutated were significantly more conserved than residues that did not alter protein expression when mutated (*Figure 2C*), suggesting that the C-terminal region is important in regulating or maintaining protein stability.

## MAGEA variants are degraded through the ubiquitin proteasome pathway

We speculated that the low expression of the cancer-associated MAGEA variants is ubiquitin proteasome dependent owing to their reported interaction with the various E3 ubiquitin ligase (*Doyle et al., 2010*). Indeed, while a subset of MAGEA variants had lower level of transcripts than wild-type MAGEA (*Figure 3—figure supplement 1*), in general their protein and mRNA expression levels did not correlate and changes in transcript levels were insufficient to explain changes in protein amounts (*Figure 3A*). More importantly, the addition of proteasome inhibitor MG132 increased protein amounts of multiple MAGEA3, MAGEA6 and MAGEA10 variants (*Figure 3B* and *Figure 3—figure supplement 2*), indicating that the proteasome is a major mediator of degradation. However, the protein amounts of a number of the MAGEA variants were not altered by MG132 treatment, suggesting that alternative mechanisms contribute to the discrepancy between RNA and protein amount in these examples. Our data demonstrated that a high number of MAGEA variants are degraded through the ubiquitin proteasome pathway. Further evaluation is needed to determine the efficacy of immunotherapy targeting these low-expression MAGEA variants in cancer.

## Cancer-specific mutations and glucose/glutamine depletion stimulate proteasome-dependent MAGEA6 degradation

Recently, an International Cancer Genome Consortium (ICGC) study listed *MAGEA6* among the top 16 significantly mutated genes across tumors in two of three computational methods used (Fisher combined p value test, p=0.07; likelihood ratio test, p=9.88E-05; convolution test, p=8.85E-05) (*Biankin et al., 2012*). The study identified two MAGEA6 mutations, $MAGEA6^{N254I}$ (tumor specimen PACA-86-T) and $MAGEA6^{H305fs*>7}$ (tumor specimen ICGC_0050_TD), in PDAC patients. To examine the effects of the cancer-specific variants in the proper cell lineage, we stably expressed the MAGEA6 variants in non-transformed human pancreatic ductal epithelial cells (HPDE-iKRAS) (*Tsang et al., 2016*) and two PDAC cell lines, BxPC-3 and MIA PaCa-2 using lentivirus (*Figure 4A* and *Figure 4—figure supplement 1*). Although quantitative real-time polymerase chain reaction (qRT-PCR) confirmed high mRNA expression of both wild-type and mutant *MAGEA6*, the protein expression of the mutants was significantly downregulated compared to wild-type. Notably, $MAGEA6^{H305fs*>7}$ loses its termination codon owing to a frameshift deletion, which allows part of the vector sequence to be translated and results in a 26–amino acid addition (ERGGRVIPSFLVQSWH YKKALLINLLQRTGHYQSK) at the C-terminus, distinct from the wild-type. This change is reflected in the size increase of the $MAGEA6^{H305fs*>7}$ protein (*Figure 4A* and *Figure 4—figure supplement 1*). The clone is referred to as $MAGEA6^{H305fs*}$ hereafter to indicate the loss of the last 11 amino acids.

Consistent with our MAGEA study in 293 T cells, addition of MG132 to the stable lines expressing MAGEA6 variants resulted in a dramatic increase in mutant protein levels, which reached levels similar to those of wild-type (*Figure 4A* and *Figure 4—figure supplement 1*), supporting proteasome-

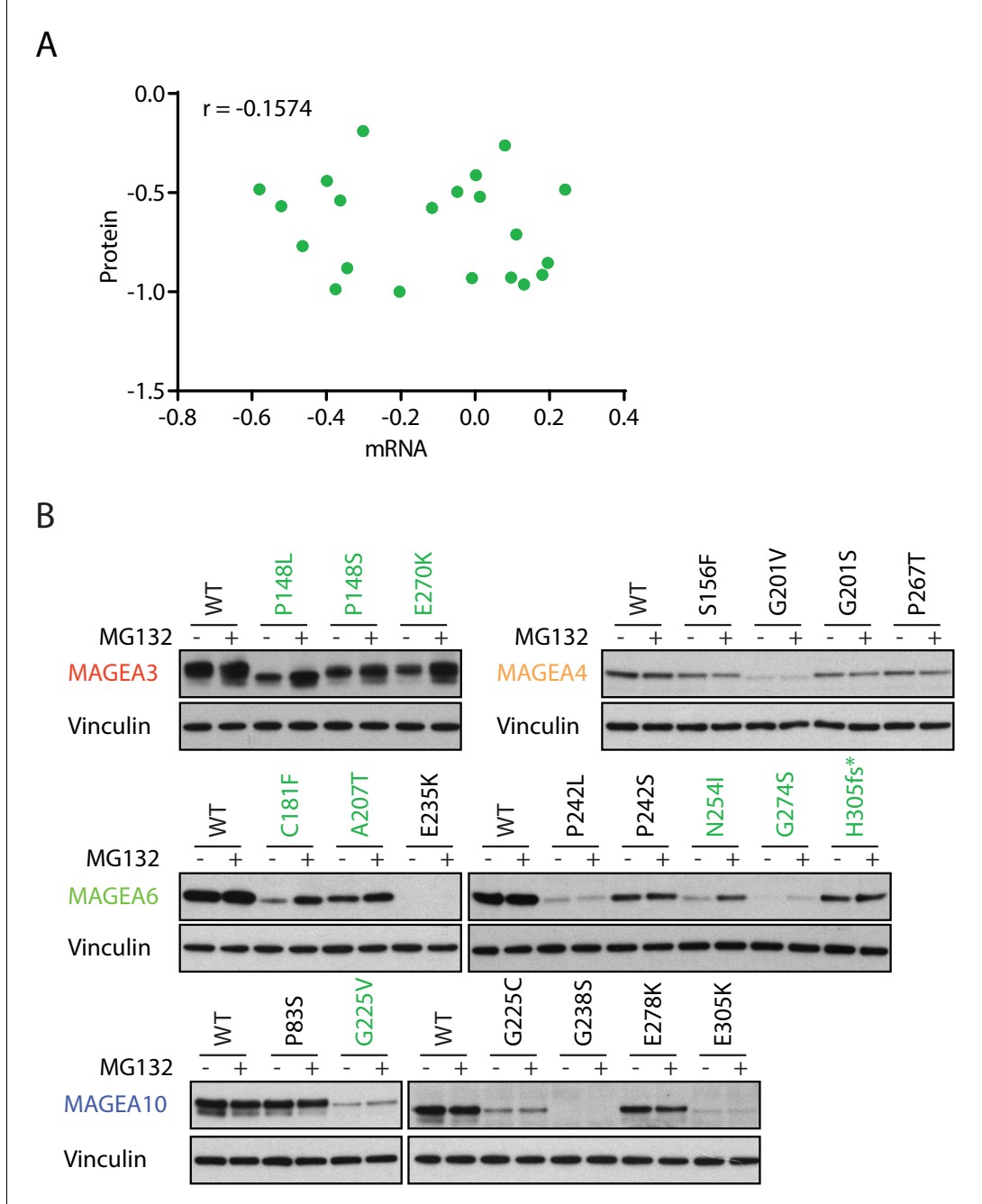

**Figure 3.** MAGEA variants are degraded through the ubiquitin proteasome pathway. (**A**) Correlation study of protein vs. mRNA expression of MAGEA variants. MAGEA variant protein and mRNA expression levels shown in *Figure 3B* and *Figure 3—figure supplement 1*, respectively, were analyzed after normalized to their corresponding wild-type. Pearson correlation coefficient r (N = 21) was calculated using GraphPad. (**B**) Immunoblot of MAGEA variants expressed in HEK293T cells with or without MG132 treatment for 9 hr. Variants that showed increased protein levels under MG132 treatment are in green.

The online version of this article includes the following figure supplement(s) for figure 3:

**Figure supplement 1.** qRT-PCR analysis of the MAGEA variants in *Figure 3B*.

**Figure supplement 2.** Densitometry analysis of protein expression of the MAGEA variants (+MG132/ -MG132) in *Figure 3B*.

mediated degradation as a means of decreasing amounts of these MAGEA6 variants. A co-immuno-precipitation assay using HEK293T cell lysate co-expressing C-terminal V5-tagged green fluorescent protein (GFP) or MAGEA6 variants and N-terminal hemagglutinin (HA) -tagged ubiquitin revealed a

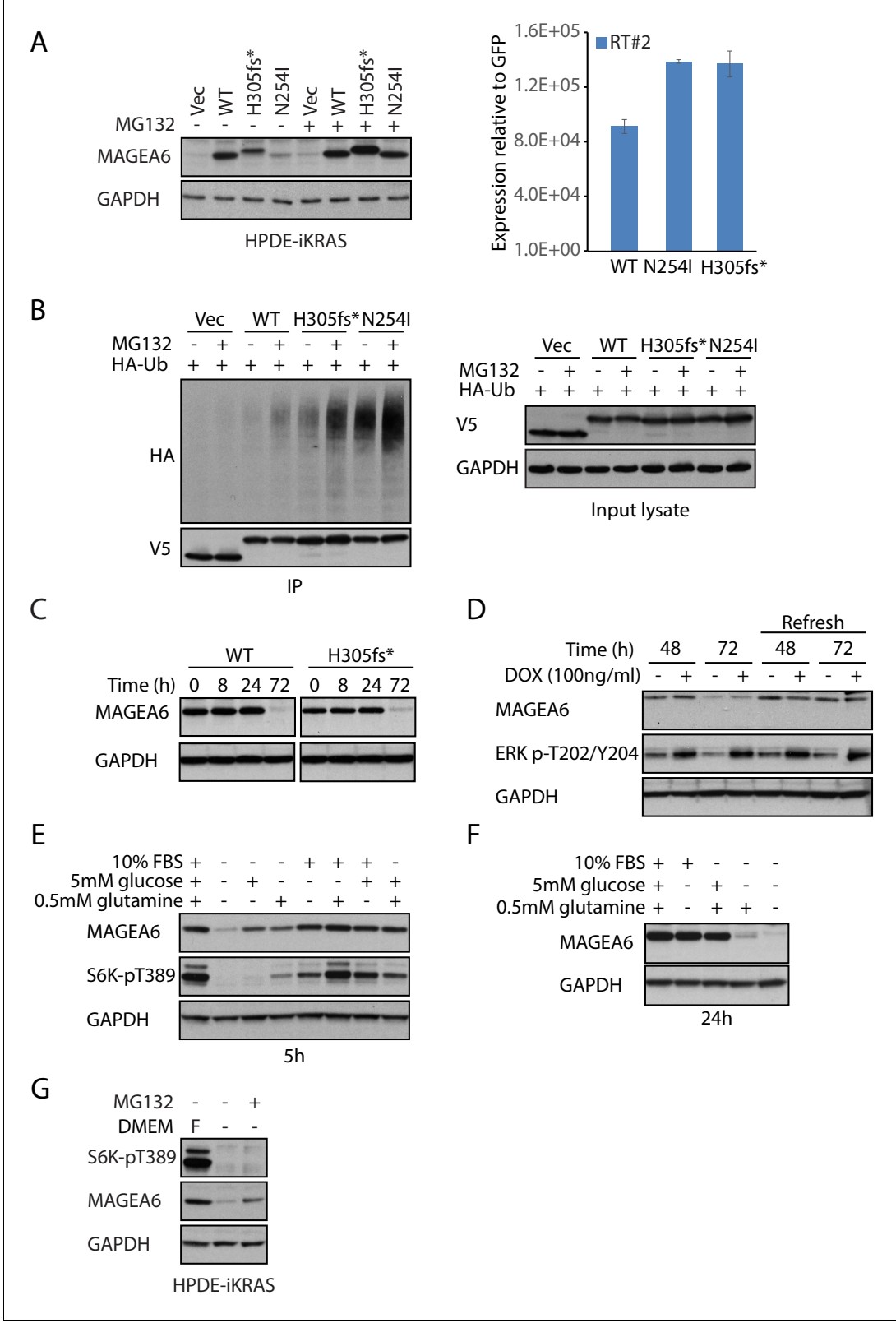

**Figure 4.** Cancer-specific mutations and glucose/glutamine depletion stimulate proteasome-dependent MAGEA6 degradation. (A) Immunoblots (left) and qRT-PCR (right) analysis of HPDE-iKRAS cells expressing GFP (Vec) and MAGEA6 variants with or without MG132. WT: wild-type, GAPDH: glyceraldehyde 3-phosphate dehydrogenase. (B) V5 pull-down assays of HEK293T lysate expressing V5-tagged GFP/MAGEA6 variants with or without MG132 treatment for 6 hr. Immunoblots of the pulled-down samples (left) and total lysate (right) are shown. HA: hemagglutinin, Ub: ubiquitin. (C)

*Figure 4 continued on next page*

*Figure 4 continued*

Immunoblot analysis of MAGEA6 in the transduced HPDE-iKRAS cells cultured in KSFM for the indicated times and (D) in the presence or absence of doxycycline at the indicated time points. KSFM was refreshed every 24 hr in the last four lanes throughout the experiment. (E) Immunoblot analysis of HPDE-iKRAS cells cultured in DMEM– supplemented with PBS, FBS, glutamine, or glucose, as indicated, for 5 hr and (F) 24 hr. (G) Immunoblot analysis of HPDE-iKRAS cells cultured in DMEM– or in FBS-, glucose-, and glucose-supplemented DMEM (F) with or without MG132 treatment for 6 hr.

The online version of this article includes the following figure supplement(s) for figure 4:

**Figure supplement 1.** Immunoblot and qRT-PCR analysis of MAGEA6 in PDAC cell models in response to MG132.

**Figure supplement 2.** V5 pull-down assays to examine endogenous polyubiquitination signal on MAGEA6 variants with or without MG132 treatment.

**Figure supplement 3.** MAGEA6 protein expression is independent of cell confluency.

**Figure supplement 4.** MAGEA6 protein expression is regulated by nutrient levels in culture media.

**Figure supplement 5.** MAGEA6 protein expression is not regulated by AKT and mTORC1 kinases.

strong HA-polyubiquitin signal on mutant MAGEA6 proteins, which was further enhanced by MG132 treatment (*Figure 4B*). We also detected a strong polyubiquitination signal on MAGEA6$^{H305fs*}$ and MAGEA6$^{N254I}$ after MG132 treatment with an antibody targeting endogenous ubiquitin in HEK293T cells (*Figure 4—figure supplement 2*), indicating that the MAGEA6 polyubiquitination observed in our co-expression study was not an artifact of ubiquitin overexpression.

While studying MAGEA6 protein stability, we observed a robust reduction of MAGEA6 variant protein levels in HPDE-iKRAS cells after prolonged culture in Keratinocyte serum-free medium (KSFM) (*Figure 4C*). We firstly examined the effect of cell confluency on MAGEA6 expression by seeding $1 \times 10^6$ MAGEA6–expressing HPDE-iKRAS cells on 10 cm and 6 cm cell culture plates. Although maximum confluency occurred on the smaller plate approximately 24 hr earlier than on the larger plate (estimated HPDE-iKRAS doubling time: 0.973 d; $R^2$: 0.9949) (*Figure 4—figure supplement 3*), we did not observe any difference between plates in MAGEA6 protein levels (*Figure 4—figure supplement 3*), challenging the argument that MAGEA6 expression is confluency dependent, at least in this model.

Next, we explored the possibility that the KRAS/MAPK pathway mediates MAGEA6 degradation, as the KRAS/MAPK pathway is hyperactivated in 95% of PDACs (*Ying et al., 2016*). We leveraged our HPDE-iKRAS cell model, which expresses *KRAS$^{G12D}$* in the presence of doxycycline (*Tsang et al., 2016*), to examine the role of KRAS/MAPK signaling in MAGEA6 expression regulation. As shown in *Figure 4D*, MAGEA6 protein remained low at 72 hr regardless of activation of KRAS/MAPK, as evidenced by the higher phosphorylation of ERK1/ERK2 Thr202 and Tyr204 (*Sebolt-Leopold, 2000*) under doxycycline treatment. Surprisingly, refreshing KSFM every 24 hr completely rescued MAGEA6 protein levels (*Figure 4D*, right four lanes), suggesting that components in the medium preserve MAGEA6 protein.

Since cancer cells often require increased amounts of nutrients, such as glucose and glutamine, to sustain the enhanced metabolic pathways required for high proliferation rates, we tested whether glucose and glutamine contributed to maintaining MAGEA6 protein levels. In vitro culture of HPDE-iKRAS *MAGEA6*-expressing cells in Dulbecco Modified Eagle Medium (DMEM) without fetal bovine serum (FBS), glucose, or glutamine (DMEM–) for 5 hr robustly downregulated MAGEA6 expression (*Figure 4E*). The reduced MAGEA6 expression was observed in the PDAC cell lines AsPC-1, BxPC-3, and MIA PaCa-2 (*Figure 4—figure supplement 4*) in DMEM–. In contrast, DMEM– supplemented with carbon sources of glucose or glutamine partially rescued MAGEA6, while DMEM– containing FBS or both carbon sources fully recovered MAGEA6 levels in both short (5 hr) (*Figure 4E*) and long (24 hr) (*Figure 4F*) term culture conditions. Along with MAGEA6 expression reduction, a reduction in p70S6K (S6K) phosphorylation was seen on residue Thr389 in DMEM– and partially supplemented DMEM– (*Figure 4E*, lanes 2, 3, and 4). S6K Thr389 is a phosphorylation site targeted by mTORC1 kinase whose activity is suppressed when nutrients are depleted (*Efeyan et al., 2012*). The rapid, highly synchronized changes in MAGEA6 expression and S6K phosphorylation status confirmed that MAGEA6 expression is nutrient sensitive. FBS-, but not glucose-dependent MAGEA3/A6 protein stability has recently been shown to be regulated by the CRL4-DCAF12 E3 ubiquitin ligase in other cell lineages including HeLa (*Ravichandran et al., 2019*). The ability of glucose and other nutrient sources to stabilize MAGEA6 protein levels in our model may reflect differences in metabolic wiring between different cell lineages due to different genomic and epigenomic backgrounds.

To examine whether MAGEA6 protein stability is maintained by mTORC1, we treated HPDE-iKRAS *MAGEA6*-expressing cells with rapamycin and MK2206 to inhibit mTORC1 or its upstream kinase AKT1, respectively. We did not observe any change in MAGEA6 protein amount after either treatment (*Figure 4—figure supplement 5*), suggesting that MAGEA6 protein levels are not regulated by the AKT/mTORC1 pathway. Consistent with a role of ubiquitination for protein degradation regulation, addition of MG132 to the starved HPDE-iKRAS cells (*Figure 4G*) and other PDAC lines (*Figure 4—figure supplement 4*) fully rescued MAGEA6 from degradation. In summary, our data show that cancer-specific mutations and carbon source depletion induce MAGEA6 degradation by proteasomes.

## Wild-type MAGEA6, but not mutant MAGEA6, suppresses basal autophagy and starvation-induced autophagy activation

In mammalian cells, mTORC1 and AMPK are key kinases that coordinate autophagy activity (*Mihaylova and Shaw, 2011*; *Zoncu et al., 2011*). mTORC1 activation promotes anabolic programs such as mRNA translation and represses catabolic programs such as autophagy when the nutrient supply is sufficient (*Bar-Peled and Sabatini, 2014*). The activation of mTORC1 can be informed by the phosphorylation status of its direct substrate S6K Thr389 (*Zoncu et al., 2011*). *AMPK*, on the other hand, is a pro-autophagy and energy-sensing kinase that is phosphorylated on Thr172 when activated (*Hardie et al., 2016*). In general, autophagy activity can be monitored by examining mTORC1 and AMPK pathways, counting LC3B puncta, and measuring autophagic flux, a frequently used technique to determine autophagy dynamics by comparing the amounts of stable protein substrates such as SQSTM1/p62 between samples in the presence and absence of lysosomal inhibitors (*Bjørkøy et al., 2005*).

In agreement with the reported autophagy inhibitory role of MAGEA3/6 (*Pineda et al., 2015*), HPDE-iKRAS, AsPC-1, and MIA PaCa-2 cells stably expressing wild-type MAGEA6 showed reduced AMPK phosphorylation on Thr172 and increased S6K phosphorylation on Thr389 (*Figure 5A* and *Figure 5—figure supplement 1*). Immunofluorescence staining of LC3B in MAGEA6–expressing HPDE cells treated with lysosome inhibitor bafilomycin A1 (BafA1) dramatically reduced the cell population with a high number of LC3B puncta (*Figure 5B*), suggesting slower autophagosome formation and thus lower basal autophagy activity levels in these cells. Autophagic flux study using BafA1 showed slower SQSTM1/p62 accumulation in MAGEA6-expressing cells compared with control cells (*Figure 5C*, left), suggesting a slower turnover of SQSTM1/p62 and therefore further supporting the anti-autophagy role of MAGEA6. Importantly, cells expressing *MAGEA6*[H305fs*] and *MAGEA6*[N254I], which are susceptible to proteasome mediated degradation, showed minimal changes in autophagy signaling (*Figure 5A*), autophagosome formation (*Figure 5B*), and autophagic flux (*Figure 5C*, right and *Figure 5—figure supplement 2*) compared with control cells.

In nutrient-deficient conditions, short-term culture of HPDE-iKRAS cells in DMEM– showed a gradual reduction in MAGEA6 expression, although MAGEA6 levels still remained sufficient to suppress AMPK Thr172 phosphorylation, reduce induction of LC3B expression (*Figure 5D*) and to impair autophagic flux, as demonstrated by the slow accumulation of SQSTM1/p62 and LC3B in the presence of BafA1 (*Figure 5E*). In contrast, cells expressing *MAGEA6*[H305fs*] and *MAGEA6*[N254I] displayed autophagy activation similar to that in control cells (*Figure 5F* and *Figure 5—figure supplement 3*). Nevertheless, long-term culture (24 hr) of cells expressing wild-type MAGEA6 in DMEM– completely abolished MAGEA6 expression and thus allowed autophagy re-initiation, as indicated by low S6K Thr389 and high AMPK Thr172 phosphorylation (*Figure 5G*). In summary, MAGEA6 suppresses both basal and short-term starvation-induced autophagy in PDAC. Its autophagy-quenching effect is revoked when MAGEA6 is downregulated either by degradation-prone mutation or by long-term starvation-induced protein degradation.

## MAGEA6 mutation status/protein amount manipulates autophagy to promote tumor progression at different stages

To investigate the oncogenic role of MAGEA6 in PDAC, we subjected non-transformed HPDE cells stably expressing wild-type MAGEA6 or GFP to bilateral subcutaneous implantation into athymic mice ($5 \times 10^5$ cells per flank; N = 5 for each cohort). In consonance with its role in autophagy suppression, MAGEA6 expression transformed HPDE cells, induced xenograft tumor growth

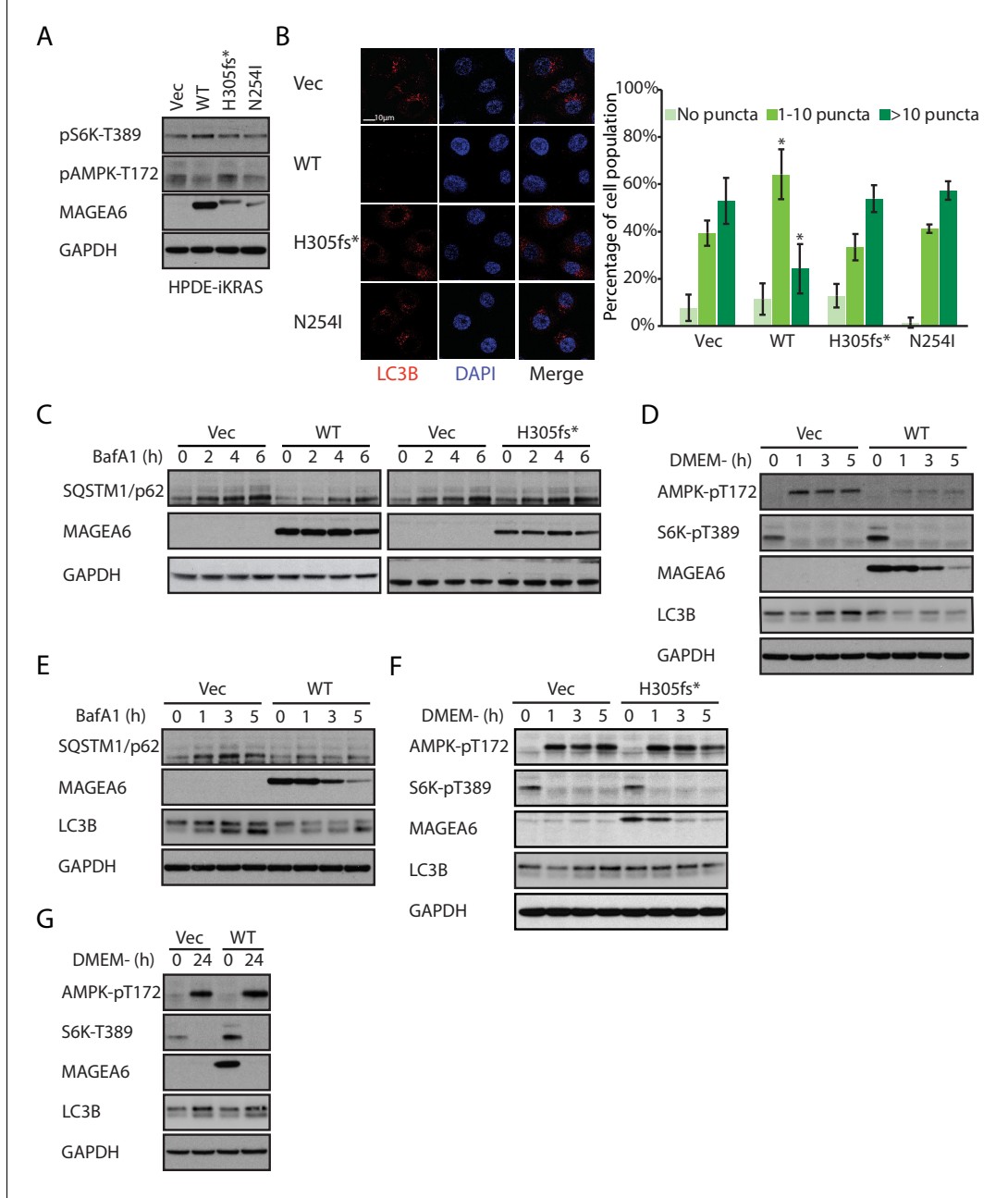

**Figure 5.** Overexpression of wild-type *MAGEA6*, but not mutant *MAGEA6*, suppresses autophagy in PDAC cell lines. (**A**) Immunoblot analysis of autophagy signaling in HPDE-iKRAS cells expressing GFP (Vec) and *MAGEA6* variants. (**B**) Immunofluorescence staining of *LC3B* puncta in the transduced HPDE-iKRAS cells. Representative photos (left) and statistical analysis (mean ± standard deviation of counted cells, N=~100 per cohort) are shown. *p=0.002; two–tailed unpaired t-test. (**C**) Immunoblot analysis of autophagy substrate SQSTM1/p62 in the transduced HPDE-iKRAS cells treated with BafA1 for the indicated time points., Immunoblot analysis of (**D**) autophagy signaling and (**E**) SQSTM1/p62 accumulation in wild-type MAGEA6 expressing and (**F**) autophagy signaling in MAGEA6^H305fs* expressing HPDE-iKRAS cells under nutrient-deficient conditions. (**G**) Immunoblot analysis of autophagy signaling in wild-type MAGEA6 expressing cells under prolonged nutrient-deficient conditions.

The online version of this article includes the following figure supplement(s) for figure 5:

**Figure supplement 1.** Immunoblot analysis of autophagy signaling of transduced AsPC-1 and MIA PaCa-2.

**Figure supplement 2.** Immunoblot analysis of the accumulation of autophagy substrate SQSTM/p62 in the transduced HPDE-iKRAS cells under BafA1 for the indicated time points.

**Figure supplement 3.** Immunoblot analysis of autophagy activity in transduced HPDE-iKRAS cells under nutrient-depleted conditions as indicated.

(*Figure 6A*), and led to poor survival (*Figure 6B*). In addition to autophagy suppression, MAGEA3/6 has been reported to enhance the degradation of the tumor suppressor p53 (*Doyle et al., 2010*), which may contribute to PDAC development. However, we did not find any association between endogenous MAGEA6 and p53 expression in HPDE-iKRAS or in six other PDAC cell lines (*Figure 6—figure supplement 1*). Furthermore, ectopic expression of *MAGEA6* variants in the BxPC-3 and MIA PaCa-2 lines did not induce changes in p53 protein levels in the presence or absence of the proteasome inhibitor MG132 (*Figure 4—figure supplement 1*), indicating that MAGEA6 drives PDAC independently of p53.

As stated earlier, autophagy exhibits both oncogenic and tumor-suppressing activities, largely depending on progression status of individual tumor. It is still unclear how autophagy is regulated to carry out these divergent activities during tumor development. The fact that MAGEA6 functions as a bona fide oncogene in PDAC but is subjected to degradation-prone mutation leads us to hypothesize that (1) MAGEA6 is an autophagy coordinator manipulating autophagy activity at different disease stages to promote tumor initiation and further development and (2) the nutrient-sensing stability of MAGEA6 provides autophagy regulation for better tumor survival under metabolically stressed conditions.

In agreement with the anti-tumorigenic role of MAGEA6 at a late disease stage, knockdown of MAGEA6 in the transformed PDAC cell line BxPC-3, which has high endogenous wild-type MAGEA6 expression, dramatically increased tumor volume compared with xenografts without MAGEA6 knockdown (*Figure 6C* top and *Figure 6—figure supplement 2*). Immunohistochemical staining of the BxPC-3 tumors revealed a robust decrease in LC3B levels and, thus, elevated autophagy activity in the *MAGEA6* knockdown cohort (*Figure 6C* bottom). In support of this contention, knockdown of *ATG7* and *VPS34*, key players for autophagy activation, suppressed tumor growth induced by *MAGEA6* downregulation, consistent with autophagy activity playing a key role in promoting tumor growth in response to MAGEA6 depletion in BxPC-3 (*Figure 6D* and *Figure 6—figure supplement 2*). We hypothesized that MAGEA6–dependent tumorigenicity is most critical at an early disease stage when low autophagy activity is preferred. We posited that the tumorigenicity of MAGEA6 and its autophagy suppression then subside as the tumor grows owing to spontaneous gene mutations and nutrient stress.

We next sought to examine the role of MAGEA6 over the course of cancer progression in patients. Since PDAC patients often present with late-stage disease and a low 5 year survival rate, we reasoned that lung cancer is a better option for evaluating the impact of MAGEA6 on different stages of cancer development (initiation and progression) because (1) a large number of patients with lung cancer showed high MAGEA6 expression and mutation events, and (2) approximately one-third of MAGEA6 mutation events (6/17) found in the ICGC lung cancer database displayed reduced protein expression (*Figure 6—figure supplement 3*). A Kaplan-Meier survival analysis of 889 patients with lung cancer showed a significant association of high *MAGEA6* expression and poor overall survival in cancers at stage I (hazard ratio = 1.96, p<0.0001) but not in those at stage II (hazard ratio = 1.1, p=0.6088) or stage III (hazard ratio = 1.06, p=0.8441) (*Figure 6E*). Collectively, our in vivo data and the patient survival analysis strongly support the hypothesis that wild-type and mutant MAGEA6 play distinct roles to promote tumor initiation and development at different disease stages.

## Discussion

Our study showed that a high number of MAGEA variants identified in pan-cancer analyses were associated with reduced protein expression, suggesting that the impact of these MAGEA gene mutations on potential and MAGEA directed cancer immunotherapy may need further evaluation. In our investigation of the relationship between MAGEA6 and autophagy, we found that intact MAGEA6 suppresses autophagy, which can be oncogenic in early-stage disease when low autophagy activity is preferred (*Kimmelman, 2011*). The degradation of MAGEA6 through ubiquitin proteasome pathway in late-stage disease revokes autophagy and further promotes cancer progression (*Figure 7*). Further study is required to examine whether TRIM28, or another E3 ligase, induces MAGEA6 degradation.

Our findings not only elucidate the regulation of tumor evolution but also reveal a novel crosstalk mechanism between autophagy and the ubiquitin proteasome system, which represent the two

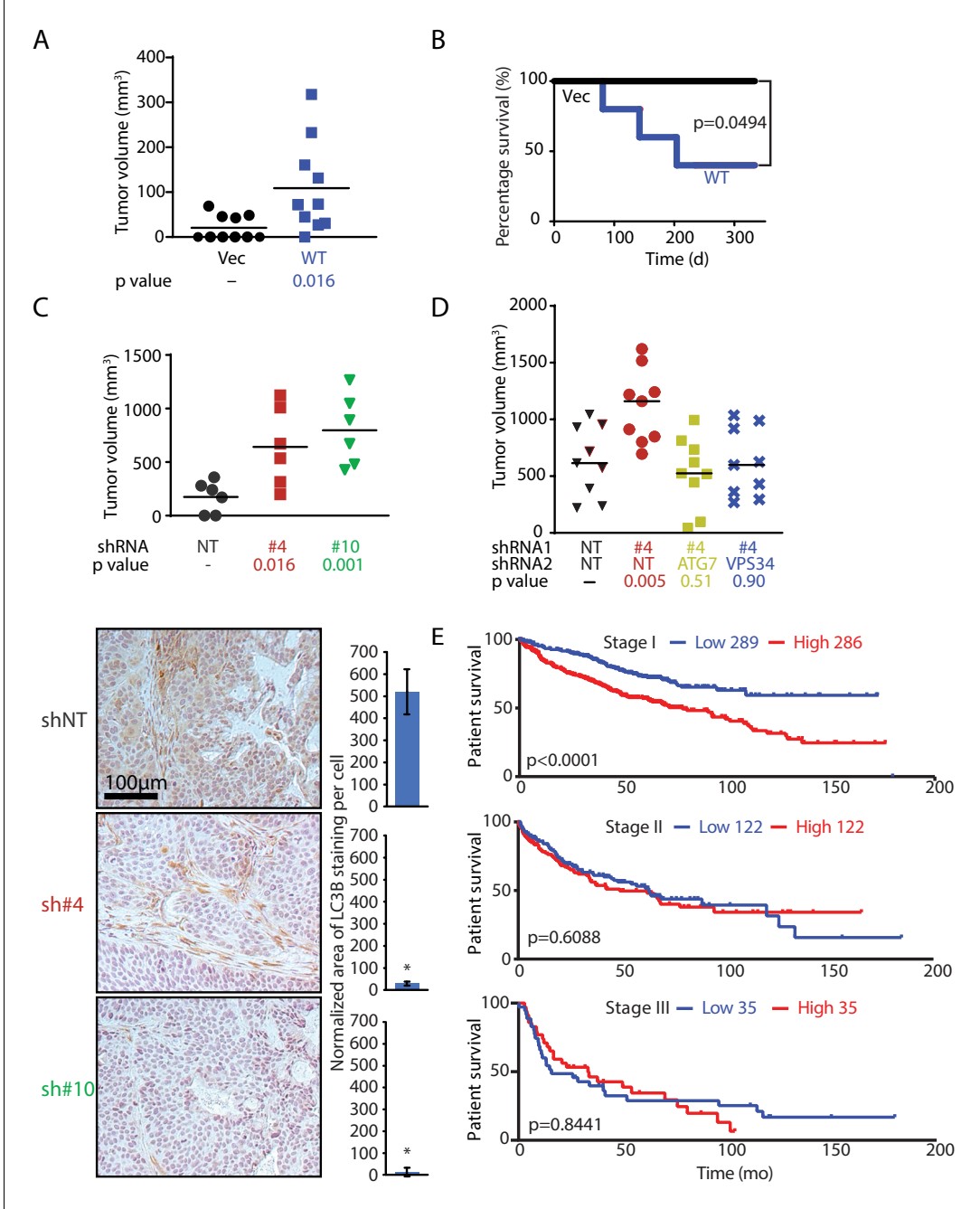

**Figure 6.** *MAGEA6* mutations and expression variation manipulate autophagy to promote tumor progression at different stages. (A) Tumor volume plot (50 d after injection) and Kaplan-Meier survival plot (B) of xenograft assay using HPDE-*iKRAS* cells expressing GFP (Vec) or MAGEA6 (five mice, two injections each, N = 10). P value was calculated by two-tailed unpaired t-test in (A) and log-rank test in (B). (C) (Top) Tumor volume plot (100 d after injection) of xenograft assay using BxPC-3 cells transduced with *MAGEA6* shRNAs (#4 and #6) and non-targeting shRNA (NT). P value was calculated by two-tailed unpaired t-test (three mice, two injections each, N = 6). (Bottom) Representative photos and quantification of *LC3B* immunohistochemical staining in the xenograft tumor samples. P value was calculated by two-tailed unpaired t-test (>150 cells analyzed per cohort). (D) Tumor volume plot (100 d after injection) of xenograft assay using BxPC-3 cells transduced with *MAGEA6 (#4), ATG7, VPS34* shRNAs and non-targeting shRNA (NT). P value was calculated by two-tailed unpaired t-test (nine mice, one injection each, N = 9). (E) Patient survival analysis of low and high *MAGEA6* expression in stage I (top), stage II (middle), and stage III (bottom) lung cancer. Number of patients analyzed per cohort is shown (*Győrffy et al., 2013*). P value was calculated by log-rank test.

The online version of this article includes the following figure supplement(s) for figure 6:

**Figure supplement 1.** *MAGEA6* expression analysis in PDAC cell line panel.

**Figure supplement 2.** Validation of *MAGEA6, ATG7* and *VPS34* knockdown in BxPC3 cells.

*Figure 6 continued on next page*

*Figure 6 continued*

**Figure supplement 3.** Gene and protein expression analysis of *MAGEA6* variants identified from ICGC lung cancer patients.

cornerstones of cellular catabolism. Indeed, several studies report evidence of crosstalk between these two catabolic pathways. For example, previous studies identified multiple protein substrates targeted by the autophagy and ubiquitin proteasome pathways (*Milani et al., 2009*; *Pandey et al., 2007*). Further analysis showed that impairment of proteasome activity led to activation of the autophagy pathway. The crosstalk between these two systems is believed to be a potential compensatory mechanism allowing cells to reduce levels of ubiquitinated substrates and provide tumorigenic advantages. Under the same theory, the reactivation of autophagy induced by low *MAGEA6* expression may provide extra intracellular catabolic capability to continuously downregulate MAGEA6 targets and sustains tumor development. Further study of this crosstalk may help elucidate the importance of cellular catabolism for tumorigenesis.

Although our study mainly focused on the tumor biology of a few *MAGEA6* variants, the MAGE family involves more than 40 genes classified as CTA genes, broadly expressed in many tumor types (*Barker and Salehi, 2002*; *Simpson et al., 2005*). Other MAGE members with similar biological properties may play a related role in cancer. For example, MAGEA2, A3, A6, and C2 were shown to bind to the TRIM28 E3 ligase with similar strength and to boost ubiquitin ligase activity against p53 (*Doyle et al., 2010*). The large number of CTA MAGEs and the high sequence homology among them may confer functional redundancy that permits ongoing positive selection toward

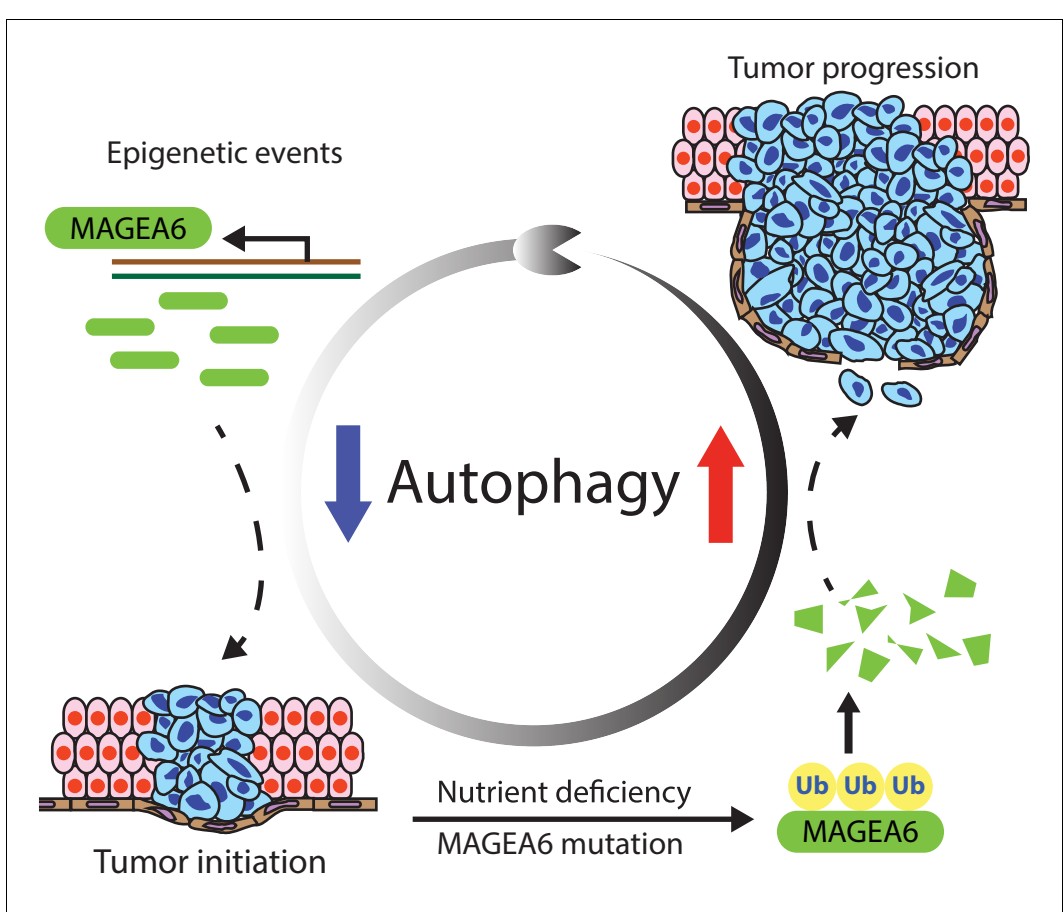

**Figure 7.** MAGEA6 drives PDAC via manipulation of autophagy. In the proposed model, *MAGEA6* expression is activated via epigenetic regulation at an early disease stage to suppress autophagy and promote tumor initiation. *MAGEA6* mutations and metabolic stress during subsequent tumor development lead to MAGEA6 degradation, reactivation of autophagy, and thus better tumor survival at the late disease stages.

diversification or acquisition of functional characteristics during tumor development. Indeed, MAGEA3 and MAGEA6 are believed to work complementarily to regulate TRIM28 E3 activity and autophagy suppression (*Pineda et al., 2015*). The co-expression of MAGEA genes and their mutation-dependent expression discrepancy may provide extra flexibility for tumors to adapt to the heterogeneity and diversity of stresses in the tumor microenvironment.

Autophagy has been studied extensively as a potential therapy target owing to its pro-tumor survival activity of conferring resistance against metabolic stress, radiation, and chemotherapies. Inhibition of autophagy using 3-methyladenine sensitizes cells to cisplatin and 5-fluorouracil in esophageal and colon cancers, respectively (*Li et al., 2009*; *Liu et al., 2011*). In PDAC cells, co-treatment of gemcitabine (the first-line therapy in the clinic) with 3-methyladenine, the AMPK inhibitor compound C, or the lysosomal inhibitor chloroquine potentiates gemcitabine-induced apoptosis (*Li et al., 2016*). Recent PDAC studies also demonstrated autophagy activation when KRAS or the downstream ERK pathway is suppressed and pharmacologic inhibition of autophagy regulators synergistically enhanced the ERK inhibitors-induced antitumor activity in KRAS-driven PDAC (*Bryant et al., 2019*; *Kinsey et al., 2019*). Our finding that autophagy suppression is induced by MAGEA6 and reactivated upon MAGEA6 degradation may inform combination chemotherapeutic strategies that could improve outcome in PDAC. For example, PDAC tumors expressing high level of MAGEA6 may be more sensitive to cytotoxic agents due to the low autophagy activity while tumors harboring MAGEA6 degradation-prone mutations may show increased dependency on autophagy, and thus by more susceptible to treatment with an autophagy inhibitor.

In summary, our findings reveal a previously unrecognized, mutation-directed regulation of MAGEA protein stability that may have important implications for maximizing the clinical utility of MAGEA–targeted immunotherapy. These findings illuminate a molecular mechanism of metabolic stress-induced degradation of an autophagy suppressor, identify a mechanism for toggling autophagy during tumorigenesis, and provide insight into proteins that function as context-dependent oncogenes or tumor suppressors.

## Materials and methods

More information of the reagents used in the study can be found in *Supplementary file 1*.

### Cell culture

All cell lines were propagated at 37°C in 5% $CO_2$ in a humidified atmosphere. HPDE cells and HPDE-iKRAS cells were cultured as described previously (*Tsang et al., 2016*). AsPC-1, BxPC-3, Capan-1, Capan-2, Panc-1, and MIA PaCa-2 were purchased from the American Type Culture Collection and cultured according to the manufacturer's recommendations. HEK293T cells were cultured in DMEM supplied with 10% FBS and 1% penicillin-streptomycin. All human cell lines were authenticated by fingerprinting using STR testing where possible and were verified to be free of mycoplasma contamination before use. In some experiments, as indicated, cells were propagated in doxycycline (100 ng/ml), 20 nM BafA1 (Selleck Chemicals), 10 μM MG132 (Sigma-Aldrich), or glucose and glutamine-free DMEM supplemented with physiologic plasma concentrations of 5 mM glucose and/or 0.5 mM L-glutamine (Corning).

### Immunoblotting, immunoprecipitation assays, and immunofluorescence staining

Cells were lysed using radioimmunoprecipitation assay buffer (Boston BioProducts) containing cOmplete, Mini Protease Inhibitor Cocktail Tablets (Sigma-Aldrich) and PhosSTOP phosphatase inhibitors (Sigma-Aldrich). Protein lysates were separated on 4–12% Bis-Tris gel (Thermo Fisher Scientific) and transferred to polyvinylidene difluoride membranes (EMD Millipore). The following antibodies were used for immunoblotting: MAGEA3 (Abcam; ab223162), MAGEA4 (Abcam; ab76177), MAGEA6 (Abcam; ab38495), MAGEA10 (Abcam; ab83557), MAGEA12 (Thermo Fisher Scientific; PA5-67682), GAPDH (glyceraldehyde 3-phosphate dehydrogenase) (Santa Cruz Biotechnology; sc-25778), phospho-ERK1/2 (Cell Signaling Technology; 9101S), phospho-AMPK (Cell Signaling Technology; 2531S), phospho-p70 S6 kinase (Cell Signaling Technology; 9205S), phospho-AKT (Cell Signaling Technology; 9271), SQSTM1/p62 (Abcam; ab56416), Tp53 (Cell Signaling Technology; 2524), V5 (Santa Cruz Biotechnology; sc-83849-R), and ubiquitin (Abcam; ab7780). All antibodies were diluted to 1:1000 in

1% bovine serum albumin. Densitometry of immunoblots was performed using ImageJ. Vinculin served as a reference gene to correct cross-sample variations. For immunoprecipitation assays, V5-tagged MAGEA6 variants or GFP was pulled down from 500 μg of transfected HEK293T cell lysate by the Protein A/G PLUS-Agarose Immunoprecipitation Reagent (Santa Cruz Biotechnology) according to the manufacturer's protocol. For immunofluorescence, transduced HPDE-iKRAS cells were seeded at 20,000 cells/well on Millicell EZ SLIDE (EMD Millipore). Cells were fixed in 2% formalin and permeabilized by 0.05% saponin/phosphate-buffered saline (PBS). LC3B antibodies (Novus Biologicals; NB-100–2220) were used to detect protein expression of LC3B puncta and immunohistochemical staining of xenograft. Slides were mounted by SlowFade Gold Antifade Mountant with DAPI (4′,6-diamidino-2-phenylindole) (Thermo Fisher Scientific) to label DNA. Cells were analyzed by confocal microscopy with a Nikon A1RS inverted laser-scanning microscope and NIS-Elements software.

## qRT-PCR

The following qRT-PCR primers were used for MAGEA6: **RT#1**: F-5′-GGAGAAAATCTGGGAGGAGC R-5′-TAGCTGGTTTCAATGAGGGC, **RT#2**: F-5′-GAGGACTCCAGCAACCAAGA R-5′-AGTAC TGCCAATTTCCGACG. For part of the mRNA transcript shared by all MAGEA variants, we used **RT#3**: F-5′-CGTTGTGAGTTGGATAGTTGTGGAAA R-5′-CTTCTGGGCATCCTTCAGCC. For the reference gene, ribosomal protein L32 (RL32), we used **RT#4**: F-5′-CCTTGTGAAGCCCAAGATCG R-5′-TGCCGGATGAACTTCTTGGT. For ATG7, we used **RT#5**: F-5′-CAGTTTGCCCCTTTTAGTAGTGC R-5′-CCAGCCGATACTCGTTCAGC. For VPS34, we used **RT#6**: F-5′-CCTGGAAGACCCAATG TTGAAG R-5′-CGGGACCATACACATCCCAT. Gene- and variant-specific qRT-PCR primers used are indicated in the figures and figure legends. **RT#4** was used to correct cross-sample variations in all qRT-PCR experiments. qRT-PCR was performed in technical replicates, N = 3, as previously described (*Lu et al., 2017*).

## Cell assays

For cell proliferation assays, HPDE-iKRAS cells were plated onto white opaque 96-well microplates in quadruplicate. Cell density was assayed by CellTiter-Glo (Promega) using a Wallac Victor2 Multilabel Counter (PerkinElmer) at multiple the indicated time intervals. All data were assessed by exponential growth curve fitting analysis using Prism 4 (GraphPad).

## Animal studies

Animal studies were conducted in accordance with an approved Institutional Animal Care and Use Committee protocol at Baylor College of Medicine. HPDE-*iKRAS*[G12D] cells transduced with GFP or MAGEA6 lentivirus (in pLenti6.3 backbone) or BxPC-3 cells transduced with shRNAs were subcutaneously injected $1 \times 10^6$ (*Figure 6A, B and C*) into both flanks and $2 \times 10^6$ (*Figure 6D*) cells per site into single flank of female nude animals (Harlan) following suspension in Hank's Balanced Salt Solution at 1:1 with Matrigel (BD Biosciences). The following MISSION pLKO shRNAs were used to knock down *MAGEA6*: shRNA #4: TRCN0000151826 and shRNA #10: TRCN0000155661. For ATG7 and VPS34 knockdown, we used MISSION pLKO shRNAs: TRCN0000007584 and TRCN0000196247 respectively. The slides were scanned by a Pannoramic 250 Flash whole slide digital scanner (3DHISTECH).

## Immunohistochemistry

Murine lung tissues or subcutaneous tumors were excised, washed in PBS, and fixed in formalin for 48 hr. Once fixed, samples were dehydrated in graded ethanol series followed by xylene. Samples were then embedded in paraffin, sectioned onto slides (5 mm thick), and allowed to dry. For immunohistochemical staining, slides were deparaffinized using a standard procedure (xylene for two washes, 100% ethanol for two washes, 95% ethanol, 70% ethanol, and 50% ethanol). Slides were then processed using VECTASTAIN Elite ABC HRP Kit and DAB Peroxidase Substrate Kit (Vector Laboratories, PK-6101, SK-4100). Briefly, after the deparaffinizing procedure (described above), antigen retrieval was completed using 0.01 M sodium citrate buffer (pH 6.0), for 15 min at 95℃, followed by blocking with 0.3% $H_2O_2$ for 30 min. Slides were stained for LC3B antibody (1:4000, Novus Biologicals; NB100-2220) and processed using the VECTASTAIN and DAB kits in accordance with

the manufacturer's instructions. LC3B staining was quantified using Immunohistochemistry (IHC) Image Analysis Toolbox on ImageJ following the instructions from the plugin developers.

## Evolutionary conservation study

Conservation of protein sequences was analyzed via the ConSurf (http://consurf.tau.ac.il/); (*Ashkenazy et al., 2016*) online server. For proteins with a known 3D experimental structure, we used the known 3D structures as the starting point. Otherwise, we used protein sequences as input. ConSurf performed Basic Local Alignment Search Tool to collect homologous sequences with default parameters. Normalized conservation scores were reported for each protein sequence.

## Kaplan-Meier survival analysis

Lung cancer patient survival was analyzed using KM plotter (*Győrffy et al., 2013*) with MAGEA6-specific probe: 214612_x_at. Patients with high and low MAGEA6 expression were divided by population median.

## Acknowledgements

This work was supported by the Cancer Prevention and Research Institute of Texas (CPRIT; RP120046), Lustgarten Foundation (RFP-B-042), the 2014 Pancreatic Cancer Action Network AACR Career Development Award (14-20-25-SCOT) funding to KLS, and the National Institute of Health (NIH) R01 (5R01CA211176-04) funding to KLS and SHE. This work was also supported by the NIH grants (U01CA168394 to KLS and GBM; R01CA175486 to H Liang, and U24209851 to H Liang and GBM). TD was supported by a training grant from The Cullen Foundation. HLu was supported by a training grant from CPRIT (RP140102). KK was supported by an NIH/National Cancer Institute fellowship (1F32CA221015-01). The University of Texas MD Anderson Cancer Center is supported in part by the NIH through Cancer Center Support Grant P3016672.

The results shown here are in part based upon data generated by the TCGA Research Network: https://www.cancer.gov/tcga.

## Additional information

### Funding

| Funder | Grant reference number | Author |
|---|---|---|
| National Institutes of Health | 5R01CA211176-04 | Sarah H Elsea |
| Cancer Prevention and Research Institute of Texas | RP120046 | Kenneth L Scott |
| American Association for Cancer Research | 14-20-25-SCOT | Kenneth L Scott |
| National Institutes of Health | U01CA168394 | Gordon B Mills<br>Kenneth L Scott |
| National Institutes of Health | 1103109301 | Kenneth L Scott |
| Lustgarten Foundation | RFP-B-042 | Kenneth L Scott |
| National Institutes of Health | R01CA175486 | Han Liang |
| National Institutes of Health | U24209851 | Han Liang<br>Gordon B Mills |
| Cullen Foundation | Training grant | Turgut Dogruluk |
| Cancer Prevention and Research Institute of Texas | RP140102 | Hengyu Lu |
| National Institutes of Health | 1F32CA221015-01 | Kathleen Kong |

The funders had no role in study design, data collection and interpretation, or the decision to submit the work for publication.

## Author contributions
Yiu Huen Tsang, Conceptualization, Data curation, Formal analysis, Supervision, Validation, Investigation, Visualization, Methodology, Project administration; Yumeng Wang, Data curation, Formal analysis, Methodology; Kathleen Kong, Data curation, Investigation, Visualization; Caitlin Grzeskowiak, Formal analysis; Oksana Zagorodna, Turgut Dogruluk, Hengyu Lu, Nicole Villafane, Venkata Hemanjani Bhavana, Daniela Moreno, Investigation; Sarah H Elsea, Resources; Han Liang, Supervision, Funding acquisition; Gordon B Mills, Kenneth L Scott, Conceptualization, Resources, Supervision, Funding acquisition

## Author ORCIDs
Yiu Huen Tsang  https://orcid.org/0000-0002-0895-6854
Turgut Dogruluk  http://orcid.org/0000-0002-9212-9471

## Ethics
Animal experimentation: This study was performed in strict accordance with the recommendations in the Guide for the Care and Use of Laboratory Animals of the National Institutes of Health. All of the animals were handled according to approved institutional animal care and use committee (IACUC) protocols (AN-5428) of Baylor College of Medicine. The protocol was approved by the Committee on the Ethics of Animal Experiments of Baylor College of Medicine. All surgery was performed under isoflurane anesthesia, and every effort was made to minimize suffering.

## Decision letter and Author response
Decision letter https://doi.org/10.7554/eLife.48963.sa1
Author response https://doi.org/10.7554/eLife.48963.sa2

# Additional files
## Supplementary files
• Supplementary file 1. Key Resources Table.

• Supplementary file 2. List of clinical trials of cancer immunotherapy targeting MAGEA antigens. Clinical trials of MAGEA-targeted immunotherapy reported from clinicaltrials.gov.

• Supplementary file 3. MAGEA variants reported in skin, lung and pancreas cancer patients. Recurrent mutations of MAGEA3, A4, A6, A10 and A12 genes identified from skin, lung and pancreas cancer patients in ICGC, TCGA and COSMIC databases.

• Transparent reporting form

## Data availability
All data generated or analysed during this study are included in the manuscript and supporting files. Source data files have been provided for Figures 1.

The following datasets were generated:

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
