## [Decision Letter]

**Acceptance summary:**

The findings provide mechanistic insight into the significance of MAGE-A6 mutations in cancer. These important findings have implications for understanding treatment and progression of pancreatic cancer, a disease in need of new treatment approaches based on a thorough understanding of cancer biology.

**Decision letter after peer review:**

Thank you for submitting your article "Differential expression of MAGE-A6 toggles autophagy to promote pancreatic cancer progression" for consideration by *eLife*. Your article has been reviewed by three peer reviewers, including a Reviewing Editor, and the evaluation has been overseen by Jeffrey Settleman as the Senior Editor. The reviewers have opted to remain anonymous.

The reviewers have discussed the reviews with one another and the Reviewing Editor has drafted this decision to help you prepare a revised submission.

The authors examine MAGE-A6 that is highly mutated in a number of cancers and show that MAGE-A6 suppresses autophagy leading to pancreatic cancer development while cancer specific variants and nutrient deprivation stimulate its proteasome-dependent degradation and hence autophagy reactivation thereby promoting tumour progression that reflect the paradoxical role of autophagy in cancer.

Summary:

The authors examine whether mutation of MAGEA6 results in differential regulation of autophagy during tumorigenesis. The authors show that: 1) MAGE genes are mutated in tumors, 2) Some mutations lead to protein instability and a portion are degraded by the proteasome, 3) MAGEA6 protein stability is regulated by nutrient deprivation, 4) MAGEA6 regulates autophagy, 5) Differential tumor promoting and suppressing activity of MAGEA6, and 6) expression of MAGE-A6 in lung cancer and show a relationship between expression and survival in stage 1 but not stage 2 and 3 disease. The novelty of the paper is diminished by previous work that has established MAGEA6 connection to autophagy and degradation by nutrient deprivation. The conceptual advance in this work relates to the idea that mutation of MAGEA6 may allow tumors to tune autophagy regulation such that during initiation MAGEA6 levels are high and thus repress autophagy, while in later stages MAGEA6 levels are decreased due to nutrient limitations and thus autophagy is derepressed. This idea is novel and potentially exciting.

Essential Revisions:

1) The studies in Figure 6 of the differential tumor promoting and suppressing activity of MAG-A6 are indeed tantalizing. Nonetheless it is necessary to show the changes on tumourigenesis are indeed due to regulation of autophagy by genetic ± pharmacological means. For example by knocking down a component of the Beclin complex or pharmacological inhibition of vps34 in combination with atg7 knockdown.

2) Additional markers of autophagy are required for Figure 5. The authors need to use other methods beside p-AMPK and p-S6K as indicators of autophagy in Figure 5D-G. For example experiments looking at autophagosome, LC3 conversion, and p62 protein levels upon nutrient deprivation in cells expressing MAGEA6 wild-type or mutant should be performed.

3) Please provide further details and justification around how high and low expression of MAGE-A6 are 'statistically' derived in Figure 6F.

---

## [Author Response]

Essential Revisions:1) The studies in Figure 6 of the differential tumor promoting and suppressing activity of MAG-A6 are indeed tantalizing. Nonetheless it is necessary to show the changes on tumourigenesis are indeed due to regulation of autophagy by genetic ± pharmacological means. For example by knocking down a component of the Beclin complex or pharmacological inhibition of vps34 in combination with atg7 knockdown.

We appreciated the suggestion and agreed with the importance to confirm the role of autophagy for tumor growth when MAGE-A6 is depleted. We performed knockdown of *MAGE-A6* alone or together with key players required for autophagy activation and performed in vivo tumor growth assay with increased mice number per cohort as suggested by the reviewers. Our data confirmed the tumor promoting activity of MAGE-A6 downregulation and demonstrated the importance of autophagy in supporting the enhanced tumor growth induced by MAGE-A6 knockdown (Figure 6D).

2) Additional markers of autophagy are required for Figure 5. The authors need to use other methods beside p-AMPK and p-S6K as indicators of autophagy in Figure 5D-G. For example experiments looking at autophagosome, LC3 conversion, and p62 protein levels upon nutrient deprivation in cells expressing MAGEA6 wild-type or mutant should be performed.

As requested by the reviewers we included additional autophagy markers to examine the change of autophagy activity. We observed reduced LC3B protein levels under short-term starvation in cells expressing wild-type MAGE-A6 without BafA1 (Figure 5D) and a slower LC3B and p62 accumulation in the presence of autophagy inhibitor (Figure 5E), indicating that autophagy is suppressed by wild-type MAGE-A6. For the destabilised MAGE-A6 variants, we did not observe such reduction in LC3B protein level (Figure 5F and Figure 5—figure supplement 3), further supporting the argument that these MAGE-A6 variants do not actively regulate autophagy.

3) Please provide further details and justification around how high and low expression of MAGE-A6 are 'statistically' derived in Figure 6F.

As requested, we have clarified the definition for high and low MAGE-A6 expression in the study of Figure 6E. We divided the lung cancer patients by population median of MAGE-A6 expression. We added a paragraph to describe the study in the Materials and methods section.